# Adaptation of Proteome and Metabolism in Different Haplotypes of *Rhodosporidium toruloides* during Cu(I) and Cu(II) Stress

**DOI:** 10.3390/microorganisms11030553

**Published:** 2023-02-22

**Authors:** Philipp Cavelius, Selina Engelhart-Straub, Alexander Biewald, Martina Haack, Dania Awad, Thomas Brueck, Norbert Mehlmer

**Affiliations:** TUM School of Natural Sciences, Technical University of Munich (TUM), 85748 Garching, Germany

**Keywords:** *Rhodosporidium*, copper stress, stress response, fatty acids, carotenoids, time-resolved proteomics

## Abstract

*Rhodosporidium toruloides* is a carotenogenic, oleogenic yeast that is able to grow in diverse environments. In this study, the proteomic and metabolic responses to copper stress in the two haplotypes IFO0559 and IFO0880 were assessed. 0.5 mM Cu(I) extended the lag phase of both strains significantly, while only a small effect was observed for Cu(II) treatment. Other carotenogenic yeasts such as *Rhodotorula mucilaginosa* are known to accumulate high amounts of carotenoids as a response to oxidative stress, posed by excess copper ion activity. However, no significant increase in carotenoid accumulation for both haplotypes of *R. toruloides* after 144 h of 0.5 mM Cu(I) or Cu(II) stress was observed. Yet, an increase in lipid production was detected, when exposed to Cu(II), additionally, proteins related to fatty acid biosynthesis were detected in increased amounts under stress conditions. Proteomic analysis revealed that besides the activation of the enzymatic oxidative stress response, excess copper affected iron–sulfur and zinc-containing proteins and caused proteomic adaptation indicative of copper ion accumulation in the vacuole, mitochondria, and Golgi apparatus.

## 1. Introduction

In recent years, the red yeast *Rhodosporidium toruloides*, part of the Basidiomycete family, has attracted increasing interest in the scientific community [1,2,3]. Its characteristic coloration is caused by the synthesis and accumulation of different carotenoids. The yeast can accumulate up to 70% lipids of its dry weight [4,5,6,7]. Additionally, the yeast has the capability to utilize a wide range of carbohydrates as a carbon source present in low-cost biomass waste streams, including monosaccharides such as hexose and pentose, oligosaccharides such as cellobiose, sucrose, and maltose, and alcohols such as glycerol and ethanol [1,8,9]. These features are promising for future industrial applications [4,10]. There are two haplotypes within the species of *R. toruloides*, also called mating types A1 and A2. Two commonly utilized strains of *R. toruloides*, each representing one haplotype, are strains IFO0880 (A2) and IFO0559 (A1), respectively. When comparing their genomes, over one million single nucleotide variations were detected. Moreover, IFO0559 exhibited great similarities with *R. toruloides* NP11, with only 443 single nucleotide variations between their genomes [10]. Most recent studies have focused mainly on one of the two haplotypes, but comparisons between those strains are scarce.

As with all aerobic organisms, *R. toruloides* is exposed to oxidative stress, or more specifically, to the intracellular formation of reactive oxygen species (ROS), which can cause damage to proteins, lipids, and nucleic acids. Generally, cellular detoxification of these ROS compounds is mediated by specialized oxidoreductases, including peroxidases, catalases, and superoxidases [11,12]. Additionally, carotenoids are known to exhibit antioxidant activities, acting as radical quenchers that are able to inactivate different ROS [12,13]. Furthermore, both the cell wall and cell membrane act as barriers to protect the cell from the environment and undergo modification when exposed to stress [14].

The inorganic element copper can either exist in its reduced, Cu(I), or oxidized, Cu(II), state, allowing for its diverse function in cellular structure and catalysis. Cu(I) has an affinity for thiol and thioether groups, which can be found in cysteine and methionine, while Cu(II) has an affinity for coordination with oxygen and imidazole nitrogen groups, which can be found in aspartic and glutamic acid or histidine [15]. As a trace element, copper is essential for all organisms, as it enables the activity of numerous enzymes and electron transport proteins. Moreover, intracellular homeostasis is tightly regulated in order to balance the cells’ demand for copper while simultaneously avoiding excessive accumulation. In *Saccharomyces cerevisiae*, it was demonstrated that vacuoles and mitochondria are the major compartments of storage for this element [16]. It was also hypothesized that vesicles as well as lipid droplets, through their capacity to bind heavy metals to a variety of proteins, organic acids and bases, or other molecules, could be involved in the intracellular storage and transport of copper ions [16,17]. Free copper is able to subject cells to lethal amounts of oxidative stress, resulting in strong biotoxic effects, thereby damaging proteins, nucleic acids, and lipids or inhibiting enzymatic activity [15,18,19].

In this study, the stress response of the two *R. toruloides* haplotypes IFO0880 and IFO0559 to Cu(I) and Cu(II) was investigated. For this purpose, alterations in growth characteristics, lipid, and carotenoid accumulation as well as shifts in fatty acid profile were monitored. Quantitative time-resolved proteomics enabled the identification of differential levels of proteins and metabolic pathways under the copper stress conditions carried out.

## 2. Materials and Methods

### 2.1. Yeast Strain and Culture Conditions

*R. toruloides* IFO0559 and IFO0880 (corresponding to ATCC10788/CBS 14 and ATCC10657/CBS 349, respectively) were maintained on yeast extract-peptone-dextrose (YPD) agar plates (10 g L^−1^ yeast extract, 20 g L^−1^ peptone, 20 g L^−1^ glucose, 14 g L^−1^ agar). As seed cultures, 100 mL YNB liquid medium (6.7 g L^−1^ yeast-nitrogen base [without amino acids], 40 g L^−1^ glucose) in 500 mL baffled shaking flasks were inoculated with single colonies and cultivated for 48 h in a rotary incubator (New Brunswick InnovaTM 44, Eppendorf, Hamburg, Germany). All cultivations were performed in 500 mL baffled shaking flasks with Duran GL32 Membrane Vented Screw Caps (DWK Life Science, Wertheim, Germany) holding 125 mL YNB in biological triplicates. Cultures were inoculated to an OD_600nm_ of 0.2 and cultivated at 28 °C and 120 rpm. Experiments to determine growth as well as lipid and carotenoid accumulation were conducted with 0.5 mM Cu(I) and 0.5 mM Cu(II) as well as controls. For 0.5 M Cu(I), stock solution Copper(I) chloride was dissolved in 30% ammonia solution, the control was therefore also supplemented with 30% ammonia. For both conditions, final ammonia concentrations amounted to 0.03%. Cu(II) (Copper(II) dichloride) was dissolved in ddH_2_O. Samples were collected twice a day for OD_600nm_ measurements. Sampling for DCW, carotenoid titers, and fatty acid profile was performed after 48 h, 96 h, and 144 h, and for proteomic analysis after 96 h and 144 h. Cu(I), Cu(II), ammonia solution, and YNB were purchased from Carl Roth GmbH & Co. KG (Karlsruhe, Germany).

### 2.2. Growth Analysis

For growth evaluation, optical density was measured at 600 nm in a photometer (Nano Photometer NP80, IMPLEN, Munich, Germany). Standard semi-micro cuvettes made of polystyrene were used with a sample volume of 1 mL.

For dry cell weight (DCW) analysis, 10 mL of culture was sampled, followed by centrifugation (4000× *g*, 10 min) and subsequent lyophilization (−80 °C, min. 72 h). Gravimetric measurements were carried out, whereby the weight of empty vessels was subtracted from the weight of vessels containing lyophilized samples.

### 2.3. Pigment Extraction

Pigment extraction from dry biomass was carried out as previously shown [12], with the modification of larger glass beads (2 mm). In summary, cells were disrupted with glass beads, followed by pigment extraction with acetone. Absorbance was determined at 454 nm (Nano Photometer NP80, IMPLEN, Munich, Germany).

### 2.4. Fatty Acid Profile

Fatty acid analysis was carried out as previously shown [12], with the modification of using 2 mg of lyophilized biomass instead of 5 mg. Briefly, biomass was converted into fatty acid methyl ester (FAME) by MultiPurposeSampler MPS robotic (Gerstel, Linthicum Heights, MD, USA). FAMEs were analyzed by gas chromatography (GC-2025 coupled to an AOC-20i auto-injector and an AOC-20s auto-sampler, Shimadzu, Duisburg, Germany) and a flame ionization detector [20]. A Zebron ZB-wax column (Phenomenex, Aschaffenburg, Germany) was used for separation. Marine Oil FAME mix (20 components from C14:0 to C24:1; Restek GmbH, Bad Homburg, Germany) and FAME #12 mix (C13:0, C15:0, C17:0, C19:0, C21:0; Restek GmbH, Bad Homburg, Germany) were utilized as external standards, which allowed for the identification of FAMEs. Normalization was based on the internal methyl laurate (C12; Restek GmbH, Bad Homburg, Germany) standard, as previous testing indicated the absence of C12:0 in *R. toruloides* samples, allowing for comparative quantitation.

### 2.5. Proteomics

#### 2.5.1. Proteomic Analysis

Proteomics was carried out as previously shown [12]. Proteins were extracted and precipitated with Protein Extraction Reagent Type 4 (Sigma-Aldrich, St. Louis, MO, USA) (1:3, *v*/*v*) and 20% trichloric acid (*v*/*v*), followed by in-gel digestion with trypsin. Peptides were analyzed using LC-MS/MS analysis with a timsTOF Pro mass spectrometer equipped with a NanoElute LC System (Bruker Daltonik GmbH, Bremen, Germany) on an Aurora column (250 × 0.075 mm, 1.6 μm; IonOpticks, Hanover St., Rozelle, NSW, Australia) [21,22]. For every condition, three biological and two technical replicates were prepared.

#### 2.5.2. Bioinformatics Analysis

PEAKS Studio software 10.6 (Bioinformatics Solutions Inc., Waterloo, ON, Canada) [23,24,25] was used for peptide and subsequent protein identification. *R. toruloides* IFO0559 and IFO0880 protein (fasta) databases were obtained from UniProt (IFO0559: https://www.uniprot.org/proteomes/UP000016926, (accessed on 21 November 2022), 8138 proteins and IFO0880: https://www.uniprot.org/proteomes/UP000239560, (accessed on 21 November 2022), 8475 proteins). As search parameters, a precursor mass of 25 ppm using monoisotopic mass and a fragment ion of 0.05 Da were selected. Furthermore, settings included trypsin as a digestion enzyme, a maximum of two missed cleavages per peptide, FDR at 1.0%, and at least 1 unique peptide per protein was required for identification. Different experimental conditions were compared with the use of the Quantification tool PEAKSQ. Here, a mass error tolerance of 20.0 ppm, ion mobility tolerance of 0.05 Da, and a retention time shift tolerance of 6 min (Auto Detect) were selected. Only proteins with a fold change and significance of at least 2 were exported for further analysis.

KOALA (KEGG Orthology And Links Annotation, https://www.kegg.jp/blastkoala/ (accessed on 18 December 2022)) was utilized as a tool for the functional characterization of exported protein sequences [26]. Furthermore, annotations were manually validated with the NCBI and UniProt databases.

## 3. Results and Discussion

### 3.1. Influence of Cu(I) and Cu(II) on R. toruloides

Throughout the cultivation period of *R. toruloides* IFO0559 and IFO0880, the influences of Cu(I) and Cu(II) resulted in distinct differences in growth patterns (Figure 1, Figure 2 and Figure 3). As Cu(I) is not soluble in water, it was applied in a solution of 30% ammonia. The respective control group was also supplemented with 30% ammonia, resulting in a concentration of 0.03% ammonia in both groups. The control group for Cu(II) was solely YNB without any supplements (0 mM) as Cu(II) was fully soluble in water. Molar concentrations of 0.5 mM Cu(II) were chosen, as clear changes in coloration were observed in IFO0880 without severe delay or impairment of culture growth (Figure 1). As copper commonly exhibits oxidative stress, which is known to increase carotenoid content in several carotenogenic species [12,27,28], growth (OD_600nm_ and biomass) and relative carotenoid content were analyzed. Additionally, it was reported that various stress conditions prompt adaptation of the cellular membrane, including shifts in fatty acid composition. To evaluate whether such changes occur in *R. toruloides* in response to copper, growth conditions were chosen that do not result in high intracellular lipid accumulation. High lipid accumulation could conceal changes in membrane lipids, therefore hampering the analysis of such adaptations.

### 3.2. Influence of Cu(I) and Cu(II) on R. toruloides IFO0559

During the first 48 h of IFO0559 cultivation, control cultures supplemented with 0.03% ammonia grew rapidly, then growth stagnated for the next 48 h before returning to rapid growth, reaching a final OD_600nm_ of 8.51 after 148 h, while control cultures without any additives exhibited a maximum OD_600nm_ of 6.67. The addition of 0.5 mM Cu(I) had the strongest effects on growth with an extended lag phase of roughly 24 h as well as a final OD_600nm_ of only 4.37 (Figure 2a). The highest OD_600nm_ values were documented for Cu(II) supplemented cultures with a final average OD_600nm_ of 8.75. Similar results are reflected in the dry cell weight (DCW). The highest DCW of 4.06 g L^−1^ was recorded for control cultures supplemented with only 0.03% ammonia, followed by samples supplemented with Cu(II) with a respective DCW of 3.65 g L^−1^. The control group without any supplementation accumulated an average of 3.07 g L^−1^. Accordingly, the lowest DCW was obtained from cultures supplemented with 0.5 mM Cu(I) with 3.0 g L^−1^ (Figure 2b). The significant effect of Cu(I) on growth can be seen most prominently in DCW after 48 h, reaching an average DCW of 1.32 g L^−1^, whereas its ammonia control reached 3.13 g L^−1^ (*** *p* < 0.001).

Grown without any additives, IFO0559 exhibited a lipid content of 7.6% [g g^−1^_DCW_] after 96 h, which is lower than at 48 h. The reason for that could be an increase in biomass, without or with a low increase in lipid production, resulting in a lower relative lipid titer, although absolute amounts could be the same or even higher. Lipid degradation at this point, with titers increasing again to 14.8% at 144 h, is rather unlikely. Lipid content increased to 13.8% and 15.6% when the yeast was supplemented with 0.5 mM Cu(I) and Cu(II), respectively. After another two days (144 h), cultures treated with 0.5 mM Cu(II) accumulated 17.0% lipids. Cultures treated with 0.5 mM Cu(I) exhibited a higher lipid content (16.4%) than the control group treated with ammonia (10.2%) (Figure 2b). Hence, clear differences could be observed after 96 h between the control group without supplements in comparison to each of the other experimental set-ups, showing a lower overall lipid content in those controls. It is therefore likely, that 0.5 mM Cu(II) as well as 0.03% ammonia, induced early lipid biosynthesis without increasing maximum lipid accumulation capacities. Interestingly, Cu(I), though limiting growth, shows no negative effect on lipid accumulation, even exceeding lipid amounts of its ammonia control after 144 h.

As several studies noted an increase in carotenoid content in Rhodotorula species when exposed to copper [27,29,30,31], the absorbance of extracted carotenoids was measured and normalized to DCW. All cultures exhibited an increase in carotenoid content over time. Only carotenoid content of Cu(I) treated samples was reduced between 48 h and 96 h. However, after 144 h, Cu(I) sample carotenoid content exceeded those of its respective controls although not significantly (*p* = 0.188). After 144 h, the highest amounts of carotenoids were detected in the control cultures without any additives (0.14 Abs_454nm_ mg^−1^_DCW_), and no significant differences between the different sample groups were observed (Figure 2c).

**Figure 2 microorganisms-11-00553-f002:**
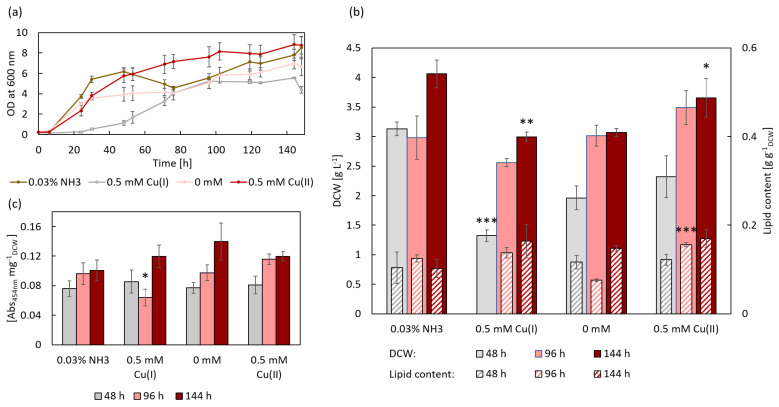
Growth, lipid, and carotenoid accumulation of *R. toruloides* IFO0559 supplemented with 0.5 mM of Cu(I) and Cu(II), as well as respective control, whereas NH_3_ served as control for Cu(I)-treated samples and 0 mM served as control for Cu(II)-treated samples. * *p* < 0.05, ** *p* < 0.01, *** *p* < 0.001 (*t*-test always evaluated against respective control), n = 3. (**a**) Growth measured as OD at 600 nm over 144 h. (**b**) Biomass formation (DCW) and lipid content (normalized to DCW) at 48, 96, and 144 h. No lipid content was determined for 0.5 mM Cu(I) at 48 h, as not enough biomass had accumulated at this time point. (**c**) Carotenoid accumulation (normalized to DCW) at 48, 96, and 144 h.

### 3.3. Influence of Cu(I) and Cu(II) on R. toruloides IFO0880

The effect of Cu(I) and Cu(II) on the growth behavior and metabolism of *R. toruloides* IFO0880 was analyzed (Figure 3). Cultures with 0.5 mM Cu(I) as well as the respective control group contained 0.03% ammonia. The highest OD_600nm_ was detected for cultures supplemented with 0.03% ammonia, with an OD_600nm_ of 7.37. The growth of cultures, supplemented with 0.5 mM Cu(I), stagnated after inoculation for 53 h in a prolonged lag phase, before growing to a maximum OD_600nm_ of 5.81. When supplemented with Cu(II), the cells grew faster compared to those without supplements (0 mM) reaching final OD_600nm_ of 5.49 and 3.00, respectively (Figure 3a). The highest DCW of 3.63 g L^−1^ was achieved in cultures with 0.03% ammonia after 48 h, while cultures supplemented with 0.5 mM Cu(I) only reached a DCW of 1.37 g L^−1^, reconfirming the severe inhibition of growth by Cu(I) (*** *p* < 0.001). While the DCW of ammonia control samples remained fairly constant over time, the DCW of Cu(I) treated samples steadily increased to a final weight of 3.05 g L^−1^. In contrast, Cu(II) treatment increased biomass formation to 2.65 g L^−1^ after 96 h and 2.33 g L^−1^ after 144 h, while the control group without supplementation reached 1.70 g L^−1^ and 1.49 g L^−1^, respectively (Figure 3b).

When grown in YNB medium without any additives (0 mM), IFO0880 exhibits a lipid content of 4.6% after 96 h. Lipid content significantly increased with the addition of 0.5 mM Cu(II) to 13.5%. When treated with Cu(I), the lipid content was 8.4%, while control cultures, supplemented solely with 0.03% ammonia accumulated 25.3%. After another two days (144 h), the lipid content in the control group (0 mM) increased to 5.2%, while cultures treated with 0.5 mM Cu(II) decreased to 9.7%. Cultures treated with 0.5 mM Cu(I) exhibited a lower lipid content (12.5%) than the control group treated with ammonia (19.3%) (Figure 3b). Accordingly, significant changes in overall lipid accumulation can be noted in samples treated with ammonia, which exhibit strongly increased lipid levels, while Cu(II) also seems to affect lipid content but to a lesser extent. Cu(I) seems to counteract some of the effects ammonia has on lipid production of IFO0880, resulting in significantly lower lipid accumulation after 96 h, which is likely to correlate with the delay in growth due to Cu(I) toxicity.

When analyzing the carotenoid absorbance, the control group (0 mM) exhibited similar absorbance values over time. Cultures treated with 0.5 mM Cu(II) exhibited no significant changes in carotenoid content after 96 h and 144 h compared to the control and between time points. For cultures treated with 0.03% ammonia an increase in carotenoid content over time was detected with a maximum absorbance of 0.08 Abs_454nm_ mg^−1^_DCW_. Cultures treated with Cu(I) exhibited significantly lower carotenoid content after 144 h (0.04 Abs_454nm_ mg^−1^_DCW_) (Figure 3c).

**Figure 3 microorganisms-11-00553-f003:**
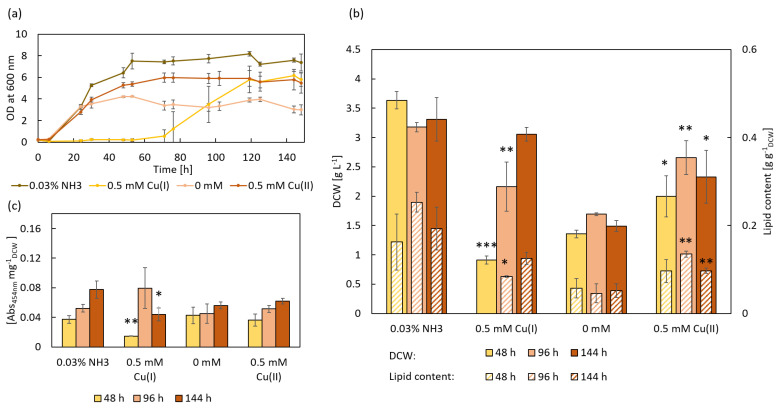
Growth, lipid, and carotenoid accumulation of *R. toruloides* IFO0880 supplemented with 0.5 mM of Cu(I) and Cu(II), as well as respective control, whereas NH_3_ served as control for Cu(I)-treated samples and 0 mM served as control for Cu(II)-treated samples. * *p* < 0.05, ** *p* < 0.01, *** *p* < 0.001 (*t*-test always evaluated against respective control), n = 3. (**a**) Growth measured as OD at 600 nm over 144 h. (**b**) Biomass formation (DCW) and lipid content (normalized to DCW) at 48, 96, and 144 h. No lipid content was determined for 0.5 mM Cu(I) at 48 h, as not enough biomass had accumulated at this time point. (**c**) Carotenoid accumulation (normalized to DCW) at 48, 96, and 144 h.

### 3.4. Comparison on Growth and Adaptation of R. toruloides IFO0559 and IFO0880

When grown in YNB medium without any additives (0 mM), IFO0559 had a higher lipid content than IFO0880 (7.6% and 4.6% after 96 h), as well as a higher DCW (3.01 g L^−1^ and 1.69 g L^−1^ after 96 h). In contrast, Zhang et al. (2016) attributed IFO0880 to a higher native capacity for lipid production than IFO0559, as total lipid amounts as well as accumulated DCW were higher. However, they also detected higher lipid content (%) in IFO0559 than in IFO0880 [10].

The supplementation of ammonia induced severe changes in IFO0880 growth behavior. For IFO0880, ammonia addition without Cu(I) resulted in the highest lipid accumulation along with an OD_600nm_ and DCW comparable to those of IFO0559. For IFO0559, growth (OD_600nm_ and DCW) as well as lipid titers were comparable to cultures grown without ammonia supplementation, although resulting in the highest overall DCW of 4.06 g L^−1^. In both haplotypes, the addition of 0.5 mM Cu(I) resulted in a prolonged lag phase (24 h and 53 h, respectively), exhibiting a higher impact on IFO0880 growth.

Additionally, both strains accumulated higher lipid titers under Cu(II) stress. The presence of copper ions enhanced lipid droplet accumulation in *S. cerevisiae*, indicating a positive correlation between lipid droplet biogenesis and accumulation of copper ions, as lipid droplets could provide storage pools for these compounds. Furthermore, the droplets might enhance the trafficking of copper to compartments such as mitochondria, vacuoles, and the Golgi apparatus [16]. The exposure to sublethal amounts of Cu(II) induced oxidative stress comparable to stress induced by ROS in *S. cerevisiae* [32]. Interestingly, supplementation with ammonia also increased the lipid titers. In IFO0559, the addition of ammonia led to similar lipid titers as the supplementation of ammonia and 0.5 mM Cu(I) after 96 h. However, in IFO0880, the sole supplementation of ammonia increased lipid amounts more than the addition of 0.5 mM Cu(I) and ammonia.

In both strains, severe prolongation of the lag phase was observed under Cu(I) conditions, but not under Cu(II) conditions. Higher toxicity of Cu(I) ions has already demonstrated in model organisms *E. coli* and *S. cerevisiae*; however, due to fast oxidation to Cu(II) ions, toxicity decreases over time. This might also be the case in the here presented study as growth resumes after a prolonged lag phase and only a limited reduction in biomass can be detected after 144 h [33,34].

*R. toruloides* is known to produce carotenoids such as astaxanthin, ß-carotene, and torularhodin, which are of interest as high-value compounds for the food, cosmetic, and pharmaceutical industries [7]. Here, after a cultivation period of 144 h, IFO0559 produces more than double the amount of carotenoids (0.14 Abs_454nm_ mg^−1^_DCW_) compared to IFO0880 (0.05 Abs_454nm_ mg^−1^_DCW_). While the production in IFO0880 increased to 0.08 Abs_454nm_ mg^−1^_DCW_ when the media was supplemented with 0.03% ammonia, production could not compete with IFO0559. Interestingly, this finding correlates with the longer lag phase of IFO0880 compared to IFO0559 during Cu(I) treatment, as the accumulated carotenoid levels of IFO0559 at 48 h significantly exceeded those of IFO0880 (** *p* = 0.0013). As the ammonia control of IFO0880 also exhibits significantly less carotenoids than IFO0559 samples (** *p* = 0.0048), this might explain the faster adaptation of IFO0559. Carotenoids are able to quench different ROS species due to their antioxidant capacities [12]; an increase in carotenoids was hypothesized when exposed to copper stress, as it was demonstrated for *Rhodotorula mucilaginosa* [27,35] and *Rhodotorula glutinis* [18,19]. Additionally, it was shown that *R. toruloides* is able to increase carotenoid content when exposed to photodynamic stress, which usually includes oxidative stress [28]. The same was demonstrated for other carotenoid producers such as *Rhodococcus erythropolis* [12]. Despite the fact that the presence of the selected amount of copper increased cellular lipid concentration, a significant increase in carotenoid accumulation could not be detected. Therefore, it can be assumed, that the oxidative stress induced by 0.5 mM copper ions (Cu(I) as well as Cu(II)) was insufficient to trigger an increase in carotenoid production or adaptation of *R. toruloides* to this type of stress is based on different metabolic changes.

To evaluate whether ammonia itself and not a shift in pH was responsible for the significant changes observed in IFO0880 growth behavior, another experiment was set up comparing both haplotypes in YNB medium (average pH 5.5), in YNB supplemented with ammonia (average pH 7.2), and in YNB adjusted to the same pH as YNB supplemented with ammonia. Interestingly, pH adjustment exhibited negative effects on the growth of both haplotypes compared to ammonia supplementation, which prompted increased growth (Appendix A).

### 3.5. Influence of Cu(I) and Cu(II) on Fatty Acid Composition of R. toruloides IFO0559 and IFO0880

Shifts in the fatty acid profiles of the haplotypes were analyzed under copper stress conditions. The percentage changes reported are the differences in total profile percentage points between the samples compared. Thus, a 10% increase to 15%, for example, is referred to as a 5% increase. For all samples, oleic acid (C18:1), linoleic acid (C18:2), palmitic acid (C16:0), and stearic acid (C18:0) represent the major fatty acid components. When cultures of IFO0559 were exposed to 0.5 mM Cu(I), the differences to non-stressed cultures at 96 h were most apparent in the increase in C16:0 (by 5.6%), C18:0 (by 6.8%), and C18:2 (by 5.2%) as well as the simultaneous decrease in C18:1 (by 22.6%). For IFO0880, similar shifts were observed when treated with 0.5 mM Cu(I). The fatty acid amount of C16:0 (3.6%) and C18:2 (8.2%) increased; however, the decrease in C18:1 was much smaller than that of IFO0559, with only 2.1%. Additionally, C18:0 decreased by 7.5% compared to the control sample (0.03% NH_3_).

When comparing the fatty acid profile of IFO0559 between 96 h and 144 h, cultures treated with 0.5 mM Cu(I) exhibited an increase in C18:2 of 4.7% as well as a decrease in C18:0 and C18:1 of 1.9% and 4.2%, respectively. Stronger changes between 96 h and 144 h were recorded for samples treated with only ammonia. C18:1 decreased by 17.7%, while C16:0 (by 3.9%), C18:0 (by 5.3%), C18:2 (by 6.9%), and C24:0 (by 2.0%) content increased (Figure 4a). When comparing the fatty acid profile of IFO0880 at 96 h to 144 h, cultures treated with 0.5 mM Cu(I) exhibited an increase in C18:1 of 6.7%, simultaneous to a decrease in C16:0, C18:2, and C18:3 of 2.8%, 4.9% and 3.9%, respectively. Between samples taken at 96 h and 144 h, those treated with only ammonia exhibited a similar fatty acid profile at both time points. C18:2 increased by 2.9%, while C18:0 decreased by 3.0% (Figure 4b).

In general, the fatty acid profile exhibited only slight alterations in samples treated with Cu(II) compared to the control samples (0 mM) (Figure 4). When treated with 0.5 mM Cu(II), the fatty acid profile of IFO0559 increased in C16:0 (by 2.1%), C18:0 (by 1.2%), and C18:2 (by 1.1%), while it slightly decreased in C13:0 (by 1.7%) and C18:1 (by 4.2%) compared to the control culture (0 mM). The fatty acid profile of IFO0880 treated with 0.5 mM Cu(II) increased in C17:0 (by 5.4%), C18:0 (by 3.6%), and C24:0 (by 2.3%) compared to its control group, while it slightly decreased in C13:0 (by 6.9%) and C18:1 (by 3.8%).

Comparison of IFO0559 at 96 h and 144 h depicts an increase in C18:1 (by 4.6% and 6.4%) and a decrease in C18:2 (by 2.4% and 3.7%) for both the control group (0 mM) and samples supplemented with 0.5 mM Cu(II) (Figure 4a). Comparing the fatty acid profiles of IFO0880 at 96 h to 144 h, the fatty acid profile remained similar. The profile of the control group (0 mM) decreased in C13:0 (by 3.5%) over time, while that of the samples treated with 0.5 mM Cu(II) increased in C18:1 and C18:2 (by 1.2% and 2.3%) and decreased in C18:0 (by 2.2%) (Figure 4b).

**Figure 4 microorganisms-11-00553-f004:**
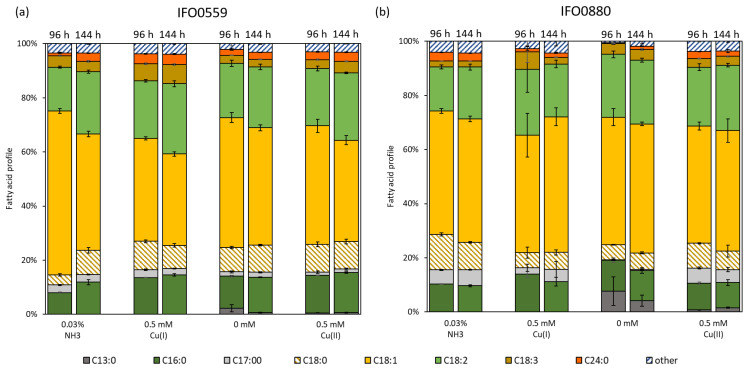
Fatty acid profiles of *R. toruloides* treated with 0.5 mM Cu(I) and Cu(II) as well as a control, whereas NH_3_ served as control to Cu(I)-treated samples and 0 mM served as control to Cu(II)-treated samples (n = 3). “Other” constitutes fatty acids with a representation below 2%, and includes, among others, C14:0, C15:0, C16:1, and C17:1. For each group, the two timepoints 96 h (left) and 144 h (right) are depicted. (**a**) Fatty acid profile of *R. toruloides* IFO0559 and (**b**) fatty acid profile of *R. toruloides* IFO0880.

Fatty acids represent an essential part of the phospholipid bilayer and are therefore major components of the cellular membrane [36]. The fluidity of a membrane can be controlled by the number of double bonds in these fatty acids and is often changed in response to stressors such as temperature changes, pH changes, and toxic compounds [37,38]. In *R. toruloides*-1588 a decrease in oleic acid (C18:1) of around 3% was observed when wood hydrolysate media was supplemented with copper sulfate [38], which is in accordance with the presented data of IFO0880 and IFO0559 in this study. In the publication of Elfeky et al., the biggest changes in the lipid profile of *R. glutinis* were observed when treated with 0.1 mM CuCl_2_, resulting in an increase in C16:0 from 13.2 to 26.4% as well as a decrease in C18:2 from 29.3 to 14.8%. Similar values were detected for 0.1 mM CuSO_4_ [31]. These drastic shifts in fatty acid profiles were not observed in this study for either IFO0559 or IFO0880.

Interestingly, differences in the fatty acid profile of IFO0559 between cultures supplemented with ammonia and cultures without supplements (0 mM) are bigger than the differences in the fatty acid profiles of IFO0880, although IFO0880 exhibited stronger alterations in growth as well as lipid titers. At time point 144 h, the fatty acid profile of IFO0559 with ammonia assimilated to the profile without supplements.

In *R. glutinis* JMT21978, it was shown that the level of oleic acid steadily decreased after glucose was exhausted from the medium. It was hypothesized that oleic acid is primarily utilized via the ß-oxidation reaction, where it might provide acetyl-CoA for the cells’ metabolism [39]. This could correlate to the decrease in C18:1 in all recorded fatty acid profiles between time points 96 h and 144 h detected in this study.

### 3.6. Effect of Cu(I) and Cu(II) on the Protein Expression

Time-resolved quantitative proteomics was performed to elucidate the cellular response of *R. toruloides* to copper stress. With this method, proteins (significance ≥ 2, fold change ≥ 2, detected in at least one sample per group, LFQ by PEAKS Studio, Appendix A) could be detected as differentially regulated at time point 96 h and 144 h in Cu(I) and Cu(II) stress. Different databases were used for IFO0559 and IFO0880, as the genomes vary between those two haplotypes. These strain-specific databases differ in protein annotation, which makes the comparison of the two haplotypes challenging as proteins can be detected under different names or are only included in one of the two databases. The highest number of differentially expressed proteins was detected in the comparison of IFO05559 at 0 mM to 0.5 mM Cu(I) after 96 h with 954 proteins, including 585 upregulated and 369 downregulated proteins. For IFO0880, the comparison of 0 and 0.5 mM Cu(II) treatment at 144 h resulted in a maximum of 946 differentially expressed proteins, with 593 upregulated and 353 downregulated proteins (Figure 5).

#### 3.6.1. Effect of Copper on Oxidative Stress

As copper toxicity is often attributed to oxidative stress [15,18], all proteomic data was analyzed for typical markers of oxidative stress (Table 1), to validate oxidative stress as a mechanism of copper toxicity in *R. toruloides* IFO0559 and IFO0880. Catalases, peroxidases, and superoxide dismutases are among the most common protein responses to oxidative stress, as they are specialized in inactivating reactive oxygen species such as peroxides and superoxides [12,19,40,41]. Additionally, the data were examined for redoxins, antioxidant enzymes, capable of interacting with respective transcription factors [19,40]. No increased amounts of proteins related to carotenoid synthesis were detected, which is in accordance with no significant increase in carotenoid titers.

Among those groups, two catalases were 2 to 3-fold downregulated, while all peroxidases, 4 different proteins in total, with the exception of glutathione peroxidase and phosphatidic acid phosphatase type 2/haloperoxidase, increased by 2 to 3-fold. Interestingly, for both haplotypes, catalase decline was only detected at 96 h (IFO0559 Cu(II) 96 h and IFO0880 Cu(I) 96 h). In contrast, peroxidases were only detected in strain IFO0559 at both time points. While the concentration of glutathione peroxidase and phosphatidic acid phosphatase type 2/haloperoxidase declined by about 2-fold in Cu(I) 144 h and Cu(II) 96 h samples, increased levels of peroxidase were observed in Cu(II) treated samples at 96 h and 144 h. Higher levels of phosphatidic acid phosphatase type 2/haloperoxidase were observed in Cu(I) samples at 144 h, as they increased 2.61-fold compared to control samples with only ammonia supplementation. In most samples, superoxide dismutase (SOD) and manganese SOD were not detected at differential levels to their control, except in IFO0880 when supplemented with Cu(II). At both time points, SOD was upregulated by about 4-fold, while manganese SOD was only detected at 144 h with about a 2-fold increase.

In the case of redoxin production, an increase in the respective redoxin amount was observed for 31 out of 41 hits (significance and fold change higher 2), with the highest increase in thioredoxin and thioredoxin h of about 6-fold (Cu(II) IFO880 96 h) and 8-fold (Cu(II) IFO880 144 h), respectively. Other types of upregulated redoxins include glutaredoxin and peroxiredoxin.

In general, more differentially regulated redoxins were detected for IFO0880 than for IFO0559; however, most downregulated redoxins were detected in samples of IFO0559 after 144 h Cu(I) supplementation followed by samples at 96 h in IFO0880 for both Cu(I) and Cu(II) treatment.

Another component in protection against oxidative stress is glutathione S-transferases (GST). Glutathione itself can act as an antioxidant and when conjugated to toxic molecules such as peroxidized lipids, resulting from ROS and their subsequent reactions with cellular components, it can act in detoxification. Moreover, GSTs act in detoxification by transporting toxins, resulting in excretion or their import into detoxification organelles such as the vacuole. In several publications, GSTs were also shown to bind free copper ions, therefore protecting cells against copper toxicity [18,42,43].

In our proteomic data set, 32 hits were identified for differentially regulated GSTs, of which 10 were detected in IFO0559 samples and 22 in IFO0880 strains. A total of 27 hits represent upregulated GST enzymes, while 5 were downregulated. Of those 5, 3 were detected in IFO0559. While Cu(II) treatment resulted in only 4 GST hits in IFO0559, Cu(I) treatment resulted in 6 hits. For IFO0880, 13 hits were detected in Cu(II) treated condition, while 9 GST hits were detected in Cu(I) samples. Significant upregulation was observed in several IFO0880 samples. In Cu(II) 96 h and Cu(I) 144 h 10- and 9.28-fold upregulation was detected. For Cu(I) 144 h, another glutathione transferase, glutathione transferase omega-1, accumulated to 12.94-fold increased levels compared to ammonia control settings. The highest accumulation of glutathione S-transferases in IFO0559 was measured for Cu(I) 96 h with 4.52-fold accumulation compared to the control.

In addition to glutathione transferases, 4 genes involved in glutathione synthesis were identified. While glutamate-cysteine ligase was detected in all IFO0880 samples, glutamate synthase was only detected in IFO0559 Cu(II) at 96 h, IFO0880 Cu(II) at 96 h, and IFO0880 Cu(I) at 96 h conditions. For these hits, upregulation between 2.86- and 6.67-fold was observed. Glutathione synthase itself was only detected in IFO0880 Cu(II) at 96 h samples with a 2-fold increase.

Glutathione-mediated detoxification of formaldehydes, a product of lipid peroxidation, is dependent on S-formylglutathione hydrolase to recycle the glutathione from the formed S-formylglutathione [44]. Other enzymes partitioning in glutathione recovery from different detoxification reactions include glutathione hydrolase, hydroxyacylglutathione hydrolase, and lactoylglutathione lyase, the latter two being part of the methylglyoxal detoxification, also known as glyoxalase I and II. With methylglyoxal being induced by oxidation of lipids (among other factors), these enzymes might be involved in the response to oxidative stress in *R. toruloides*, as was shown for different microorganisms, in which mutants defective in methylglyoxal detoxification underwent severe oxidative stress with rapid loss of viability [45,46]. All four enzymes were upregulated in different IFO0880 samples, with S-formylglutathione hydrolase showing the highest fold-change of 9.09 in Cu(II) 96 h condition.

#### 3.6.2. Effect of Copper on Iron-Sulfur Proteins

As Irazusta et al. demonstrated in their study for *R. mucilaginosa* [19], *R. toruloides* also exhibited increased levels of enzymes involved in oxidative stress response, indicating that copper toxicity in *R. toruloides* is indeed mediated through oxidative stress. However, several publications discussed an additional aspect of copper toxicity. In their work, the effect of free copper ions on iron–sulfur (Fe–S) clusters was investigated. It was postulated, that copper can inactivate iron–sulfur cluster proteins by directly damaging the iron-sulfur proteins as well as interfering with Fe–S cluster formation [47,48,49]. Especially Cu(I) competes with iron, destabilizing Fe–S cluster. Chillappagari et al. further established an increase in protein levels involved in the iron-sulfur assembly machinery under excess copper. Additionally, they found copper stress to increase the abundance of proteins involved in iron uptake [49].

In our proteomics data, 14 hits were identified for proteins of the iron–sulfur assembly machinery (Table 2). Almost all upregulated components were detected in Cu(II) samples of IFO05559 at both time points and one in Cu(I) treated IFO0880 at 144 h. In general, most hits were identified in IFO0559 (10 hits). Overall, more iron–sulfur assembly proteins were downregulated (10 hits) than upregulated (4 hits).

Furthermore, 20 protein hits involved in iron transport were identified (Table 2), of which 14 constitute upregulated proteins (2.38- to 33.33-fold). For Cu(I) and Cu(II) treatment, 11 and 9 hits were detected, respectively. All except one downregulated protein (0.44- to 0.25-fold) were detected in IFO0559, with a distribution of overall hits between IFO0559 and IFO0880 of 12 to 8. The protein with the highest and most consistent upregulation is a zip-like iron-zinc transporter, which was detected upregulated in all sample sets except for IFO0880 Cu(II), where it was not detected at all.

In summary, these results imply a measurable effect on the discussed enzymes; however, the effect is not as distinct and uniform as that described for other organisms.

**Table 2 microorganisms-11-00553-t002:** Differentially expressed proteins involved in iron-sulfur cluster assembly and iron transport discussed here. For IFO0559 and IFO0880, fold change of samples grown under Cu(I) and Cu(II) stress was compared to respective control. Protein names and UniProt accession numbers are listed. Further fold changes at 96 h and 144 h of samples are stated. Fractions below 1 depict downregulation. n.d. = not detected; up = upregulated below cut-off score; down = downregulated below cut-off score. Cut-off score: significance ≤ 2 or fold change ≤ 2.

IFO0559						IFO0880					
		Cu(I)		Cu(II)				Cu(I)		Cu(II)	
	Accession	96 h	144 h	96 h	144 h		Accession	96 h	144 h	96 h	144 h
Iron–sulfur cluster assembly											
Iron–sulfur cluster assembly protein	M7X0I3	0.42	0.44	n.d.	down	Iron–sulfur cluster assembly protein	A0A0K3C953	n.d.	up	0.28	n.d.
Iron–sulfur cluster assembly accessory protein Isa2	M7WS85	n.d.	0.47	3.70	5.44	Probable cytosolic iron-sulfur protein assembly protein 1	A0A0K3C8P4	up	n.d.	0.46	0.48
Fe–S cluster assembly protein DRE2	M7WV82	n.d.	0.24	n.d.	up	Fe–S cluster assembly protein DRE2	A0A2T0AFS9	n.d.	3.54	n.d.	up
Cytosolic Fe–S cluster assembly factor NBP35	M7WL71	up	0.28	n.d.	0.18						
Cytosolic Fe–S cluster assembly factor CFD1	M7X358	down	0.37	down	2.35						
Iron transporter											
Zip-like iron-zinc transporter	M7X8Q2	10.14	4.85	33.33	4.31	Zip-like iron-zinc transporter	A0A0K3CEY7	18.26	7.18	n.d.	n.d.
Zip-like iron-zinc transporter	M7WSK3	n.d.	0.4	up	n.d.						
Iron permease	M7X6M4	n.d.	up	0.34	up	Iron permease FTR1/Fip1/EfeU	A0A2T0A4Z9	3.94	up	n.d.	2.38
MFS transporter siderophore-iron: H+ symporter	M7WNH3	0.26	n.d.	0.30	down	MFS transporter siderochrome-iron transporter	A0A0K3C6P7	up	n.d.	n.d.	3.70
Iron/copper transporter Atx1	M7X0X4	down	0.44	up	2.81						
Iron complex transport system ATP-binding protein	M7X547	3.12	up	n.d.	down	Iron complex transport system ATP-binding protein	A0A0K3CJI3	up	2.47	down	down
Siderophore iron transporter mirC	M7X1M8	up	5.07	down	n.d.	Siderophore iron transporter mirC	A0A0K3CCV2	down	up	0.25	3.33

#### 3.6.3. Effect of Copper on Zinc Proteins

Barber et al. concluded in their review that copper also impacts zinc-dependent processes, including zinc cofactor enzymes and zinc-containing transcription factors [50]. Interestingly, both zinc finger transcription factors and zinc-containing enzymes such as zinc-type alcohol dehydrogenase were detected at differential levels compared to their controls (Appendix A). In total, 77 hits for zinc finger proteins were identified (46 in IFO0559 samples and 31 in IFO0880 samples), of which 42 are downregulated.

Additionally, proteins required for zinc homeostasis were identified (Table 3), namely cation efflux protein zinc transporter, protein of cation efflux protein family zinc transporter, zip-like iron-zinc transporter, mitochondrial zinc maintenance protein, 1 and solute carrier family 30 (Zinc transporter) member 1.

While mitochondrial zinc maintenance protein 1 is downregulated in IFO0880 Cu(II) 144 h, a 3.41-fold increase compared to the control was detected in IFO0880 Cu(I) 96 h, as well as another protein with the same designation in IFO0880 Cu(I) 144 h with a 2.06-fold upregulation.

Both the cation efflux protein zinc transporter and protein of cation efflux protein family zinc transporter were downregulated in IFO0559 during Cu(II) exposure; however, Cu(I) appears to exhibit the opposite effect increasing the production of cation efflux protein zinc transporter by 3.13-fold compared to control in IFO0559 at 144 h. Similarly to iron transporters, zip-like iron-zinc transporter was consistently upregulated in all samples except IFO0880 Cu(II).

Interestingly, downregulation higher than the cut-off (significance ≥ 2, fold change ≥ 2) of both cation efflux transporters was only detected in IFO0559, while such upregulation of mitochondrial zinc maintenance proteins was only detected in IFO0880, indicating, that both strains might have adapted differently to copper stress. However, differences might also occur through the different annotations of both strains, which would distort a direct comparison.

The presented results here are in line with the copper effects described by Barber et al. Although in a different organism, copper appears to disrupt zinc proteins, likely by displacement of zinc ions, similar to its effect on iron-sulfur clusters described above. They concluded that zinc-dependent transcriptional regulation is likely affected by copper [50], which is in accordance with our findings of zinc finger proteins being impacted by copper excess.

In the work of Hassan et al., they were able to determine that zinc stress induces copper-specific depletion, as there was no significant effect on other transition metal ions (mangan, cobalt, nickel, and iron) [51]. Interestingly, our data might indicate the opposite scenario, where copper depletes zinc likely by displacement.

**Table 3 microorganisms-11-00553-t003:** Differentially expressed proteins involved in zinc homeostasis discussed here. For IFO0559 and IFO0880, fold change of samples grown under Cu(I) and Cu(II) stress was compared to respective control. Protein names and UniProt accession numbers are listed. Further fold change at 96 h and 144 h of samples are stated. Fractions below 1 depict downregulation. n.d. = not detected; up = upregulated below cut-off score; down = downregulated below cut-off score. Cut-off score: significance ≤ 2 or fold change ≤ 2.

IFO0559						IFO0880					
		Cu(I)		Cu(II)				Cu(I)		Cu(II)	
	Accession	96 h	144 h	96 h	144 h		Accession	96 h	144 h	96 h	144 h
Zinc homeostasis											
Protein of cation efflux protein family zinc transporter	M7XE18	0.44	n.d.	0.37	0.46	Mitochondrial zinc maintenance protein 1 mitochondria	A0A0K3CHC4	up	n.d.	down	0.34
Cation efflux protein zinc transporter	M7XHN8	up	3.13	0.30	n.d.	Mitochondrial zinc maintenance protein 1 mitochondria	A0A0K3CUM1	3.41	up	up	n.d.
Solute carrier family 30 (Zinc transporter) member 1	M7WTR0	up	2.82	n.d.	down	Mitochondrial zinc maintenance protein 1 mitochondria	A0A2T0AG28	n.d.	2.06	n.d.	n.d.
Zip-like iron-zinc transporter	M7X8Q2	10.14	4.85	33.33	4.31	Zip-like iron-zinc transporter	A0A0K3CEY7	18.26	7.18	n.d.	n.d.
Zip-like iron-zinc transporter	M7WSK3	n.d.	0.4	up	n.d.						

#### 3.6.4. Storage of Copper Ions

As stated earlier, copper ions are stored in organelles in *S. cerevisiae* to avoid excessive accumulation [16]. About 25% of copper ions were stored in lipid droplets, with vacuoles and mitochondria also representing major storage compartments. Additionally, mitochondria and the Golgi apparatus are labile intracellular copper pools, which allow the cell to maintain the rapid kinetics of copper uptake and release [52]. Lipid droplets might facilitate the transport of copper ions as they interact with the Golgi apparatus, mitochondria, and vacuoles through physical contact as well as proteins present on their surface [16]. A new class of perilipin proteins was discovered in *R. toruloides* NP11, the expression of which correlates with lipid quantity, indicating their importance in lipid accumulation [53]. Fungal perilipin-like proteins were characterized to function in protecting lipid droplets from degradation [54,55]. In accordance with those publications, lipid droplet protein 1 (Perilipin-like protein) was detected in increased amounts in IFO0559 samples stressed with Cu(II) and IFO0880 samples stressed with Cu(I) and Cu(II) at 144 h.

Vacuoles are utilized as storage for copper ions. The overexpression of *hmt1,* coding for a vacuolar ABS heavy metal transporter (Htm1), in *Schizosaccharomyces pombe* enhanced the metal tolerance of the yeast. Hmt1 is associated with the vacuolar membrane and could mediate compartmentalization of heavy metals [56]. Htm1 was detected in significantly increased quantities between 2.74-fold when stressed with Cu(II) (144 h) and 17.38-fold when stressed with Cu(I) (96 h) in IFO0559 and between 3.70-fold when stressed with Cu(II) (96 h) and 9.22-fold when stressed with Cu(I) (144 h) in IFO0880 (Table 4). The fold change in quantity was higher in both strains when stressed with Cu(I). Further hint on copper compartmentalization in the vacuole is the upregulation of the vacuoler transporter chaperones 2 and 4, which both exhibited increased accumulation in IFO0559 samples treated with Cu(I), for chaperone 2 at 96 h even to a 64-fold increase. In *S. cerevisiae*, increased cytoplasmic calcium signals could also mitigate aluminum toxicity, as a mutant lacking the vacuolar calcium ion ATPase pump PMC1p exhibited a higher sensitivity to aluminum [57]. In IFO0559, a vacuolar calcium ion transporter H^+^ exchanger (M7X2U4) was detected in decreased concentrations under both Cu(I) and Cu(II) stress at 96 h, and in increased concentrations at 144 h. For cells stressed with Cu(II), a 34.66-fold increase in protein amount was detected. Additionally, a second vacuolar calcium ion transporter (M7XNE5), was increased in IFO0559 cultures stressed with Cu(I). Therefore, calcium homeostasis could similarly be able to reduce copper toxicity.

When stressed with 0.8 mM copper, a copper-resistant *thmea1* knockout mutant strain compared to the wild-type *Trichoderma harzianum* Th-33 exhibited increased expression of proteins related to the transport of copper ions into the Golgi secretory pathway. This indicates a higher number of copper ions entering the Golgi vesicles [58]. CCC2, a P-type ATPase, exports cytosolic copper ions into late or post-Golgi compartments [59]. While none of the proteins in IFO0559 could be matched to this annotation, a copper P-type ATPase in IFO0880 was upregulated 3.19-fold when cultures were stressed with Cu(I) and 2.17-fold when cultures were stressed with Cu(II) after 144 h.

Another major storage compartment for copper ions is the mitochondria [16]. These essential organelles play key roles in pathways such as ATP production, ß-oxidation, and oxidative stress clearance [60]. The mitochondrial inner membrane metallopeptidase Oma1 is part of the quality control system in mitochondria. This protein is reported to be activated by various stressors such as mitochondrial, oxidative, and heat stress [61,62,63]. Interestingly, in IFO0880 and IFO0559, Oma1 was detected in lower amounts when exposed to copper stress (Table 4). The mitochondria can exist in multiple morphologies; under oxidative stress, mitochondria could fuse into a tubular morphology. When mitochondrial fusion was blocked, sensitivity to oxidative stress was elevated in the fungus *Cryptococcus neoformans* [60]. The detected levels of maintenance of mitochondrial morphology protein 1 (MMM1), mitochondrial distribution and morphology protein 10 (MDM10), mitochondrial sensitive to high expression protein 9, and mitochondrial inner membrane protein 1 decreased when exposed to copper, with the exception of MDM10 and mitochondrial inner membrane protein 1 in IFO0880 when exposed to Cu(I). Levels of the mitochondrial outer membrane protein (IML2) and mitochondrial distribution and morphology protein 12 (MDM12) were elevated in response to Cu(II) exposure in IFO0559 and IFO0880 after 144 h, although fold changes were lower than 2.

ATM1 is part of the iron metabolism in the mitochondria of *C. neoformans,* the protein is important for the iron-sulfur cluster synthesis and heme metabolism [64]. Copper toxicity is induced through the alteration of iron-sulfur cluster homeostasis as the cofactor binding to target proteins is disrupted [65]. Sensitivity to oxidative stress was shown to be elevated in *atm1* mutants [64]. To cope with copper stress, *C. neoformans* increases the expression of ATM1, as increased export of iron–sulfur cluster precursors from the mitochondrial matrix to the cytosol, enabling iron–sulfur proteins to remain active and therefore ensure essential cellular processes [65]. The mitochondrial ABC transporter *atm* was only annotated in the genome of IFO0559, and elevated levels of the protein were detected upon exposure to copper stress (Table 4). For Cu(I) stress, a fold change of around 4 was detected, while for Cu(II) stress protein amount increased, but no significant amounts of protein were detected.

**Table 4 microorganisms-11-00553-t004:** Differentially expressed proteins involved in copper ion storage discussed here. For IFO0559 and IFO0880, fold change of samples grown under Cu(I) and Cu(II) stress was compared to respective control. Protein names and UniProt accession numbers are listed. Abbreviations are written in bold. Further fold change at 96 h and 144 h of samples are stated. Fractions below 1 depict downregulation. n.d. = not detected; up = upregulated below cut-off score; down = downregulated below cut-off score. Cut-off score: significance ≤ 2 or fold change ≤ 2.

	IFO0559					IFO0880				
		Cu(I)		Cu(II)			Cu(I)		Cu(II)	
	Accession	96 h	144 h	96 h	144 h	Accession	96 h	144 h	96 h	144 h
Copper ion storage										
Lipid droplet protein 1 (Perilipin-like protein) LDP1	M7WE51	n.d.	down	2.56	up	A0A2T0A369	down	2.23	n.d.	16.67
Vacuolar ABC heavy metal transporter Hmt1	M7XIA9	17.38	7.36	5.26	2.74	A0A0K3CL56	7.26	9.22	3.70	5.56
Vacuolar calcium ion transporter H+ exchanger	M7X2U4	0.28	2.58	0.22	34.66					
Vacuolar calcium ion transporter	M7XNE5	2.03	3.38	n.d.	n.d.	A0A2T0A7V0	n.d.	n.d.	down	n.d.
Copper P-type ATPase						A0A2T0A6Z7	2.81	3.19	n.d.	2.17
Vacuolar transporter chaperone 2	M7WMK7	64.00	17.05	n.d.	n.d.					
Vacuolar transporter chaperone 4	M7XMX5	n.d.	2.26	n.d.	n.d.					
Mitochondrial inner membrane metallopeptidase Oma1	M7X2Q6	down	n.d.	down	down	A0A0K3C9S0	down	0.46	n.d.	n.d.
Mitochondrial outer membrane protein IML2	M7X0Q4	n.d.	up	down	down	A0A2T0A031	n.d.	up	0.40	down
Maintenance of mitochondrial morphology protein 1 MMM1	M7X412	0.38	down	down	n.d.	A0A2T0AGR5	down	down	n.d.	n.d.
Mitochondrial distribution and morphology protein 12 MDM12	M7X8M4	n.d.	up	n.d.	n.d.	A0A0K3C7P5	n.d.	up	n.d.	0.50
Mitochondrial distribution and morphology protein 10 MDM10	M7WSP8	n.d.	0.34	n.d.	down	A0A0K3CJ28	0.5	down	2.33	up
Sensitive to high expression protein 9, mitochondrial	M7XMC6	0.41	n.d.	0.39	n.d.	A0A2T0AFZ7	n.d.	n.d.	n.d.	n.d.
Mitochondrial inner membrane protein 1	M7XNS0	n.d.	n.d.	0.35	down	A0A0K3CI34	0.29	down	up	n.d.
Mitochondrial ABC transporter ATM	M7XAM6	4.62	4.85	up	up					

#### 3.6.5. Effect of Copper on the Fatty Acid Profile

Zhu et al. (2012) presented a multi-omics analysis of lipid accumulation in *R. toruloides* NP11 [53], a close relative of IFO0559 [10]. The annotation of this strain was used for the proteomics analysis of IFO0559 in the present study. Proteins involved in de novo fatty acid biosynthesis of NP11 under nitrogen-limited conditions were also upregulated in our study in IFO0559, when stressed with 0.5 mM Cu(I) and Cu(II) after 96 h, including ATP citrate synthase (Acl1), malic enzyme (Me1), acetyl-CoA carboxylase (Acc1), fatty acid synthase subunit beta (Fas1), and alpha (Fas2) for fatty acid synthesis and glycerol-3-phosphate dehydrogenase (Gpd) and glycerol-3-phosphate O-acyltransferase (Gat1) for glycerolipid synthesis (Table 5). The highest fold changes were observed for Acc1 with 4.25-fold increase, and Fas2 with a 4.09-fold increase, when stressed with Cu(I). This likely correlates with the increase in lipid titers in IFO0559 when exposed to copper. The fold change of the differentially expressed proteins between Cu(I) and Cu(II) varied, with a higher fold change in proteins for cultures stressed with Cu(I), except for Me1. Moreover, Gdp and LDP1 were not detected in the cultures stressed by Cu(I). Cultures treated with Cu(I) exhibited a prolonged lag phase, and only entered the exponential phase when the respective control was already confined to the stationary phase. This could explain a stronger expression of proteins involved in de novo fatty acid biosynthesis after 96 h, as more biomass and lipid were produced in a shorter time frame. More importantly, a delta-9 fatty acid desaturase (9FAD) was increased 1.93-fold when comparing the proteome of samples treated with and without Cu(I). 9FAD synthesizes palmitoleic acid (C16:1) and oleic acid (C18:1) [66]. In contrast, the control sample with 0.03% ammonia exhibited a 22.6% higher C18:1 content than cultures treated with Cu(I) at time point 96 h, which is the highest C18:1 content recorded in all samples. Between 96 and 144 h, the oleic acid content of control samples with ammonia decreased by 17.7%. In contrast, oleic acid levels in Cu(I) samples only slightly decreased (by 4.2%). This correlates with the observed higher levels of 9FAD in Cu(I) samples.

Zhang et al. detected a two-fold increase in lipid titers in IFO0880 as a result of Acc1 and Diacylglycerol O-acyltransferase (DGA1) overexpression [10]. The overexpression of 9FAD was associated with increased lipid titers [67]. In our work, Acc1 was upregulated for cultures stressed with Cu(I), but it was downregulated when exposed to Cu(II) (Table 5). DGA1, which catalyzes the terminal step of triacylglycerol (TAG) formation, was downregulated in cultures stressed with Cu(I) and could not be detected in cultures stressed with Cu(II). Acl1 was upregulated in IFO0880 with a 2.2-fold increase. Moreover, Fas2 was overexpressed with a fold change of 2.76 for cultures stressed with Cu(I). In accordance with the lower lipid amount detected for IFO0880 when stressed with Cu(I) (Figure 3b), a decrease in the protein amounts of DGA1, Me1, Gpd, Gat1, and LDP1 was detected after 96 h. Increased lipid titers were detected for cultures stressed with Cu(II); however, only Acl1 and 9FAD upregulation was detected. Higher amounts of 9FAD could explain the increase in C18:1 for IFO0880 stressed with Cu(II) when comparing the fatty acid profiles at 96 h and 144 h. Furthermore, Gat1 and Gpd, which are responsible for glycerolipid synthesis, were detected in increased amounts.

**Table 5 microorganisms-11-00553-t005:** Differentially expressed proteins involved in fatty acid biosynthesis discussed here. For IFO0559 and IFO0880, fold change of samples grown under Cu(I) and Cu(II) stress was compared to respective control. Protein names and UniProt accession numbers are listed. Abbreviations are written in bold. Further fold change at 96 h and 144 h of samples are stated. Fractions below 1 depict downregulation. n.d. = not detected; up = upregulated below cut-off score; down = downregulated below cut-off score. Cut-off score: significance ≤ 2 or fold change ≤ 2.

	IFO0559					IFO0880				
		Cu(I)		Cu(II)			Cu(I)		Cu(II)	
	Accession	96 h	144 h	96 h	144 h	Accession	96 h	144 h	96 h	144 h
Fatty acid biosynthesis										
ATP citrate synthase Acl1	M7WHC9	2.61	up	up	3.33	A0A0K3CJ29	2.2	up	up	up
Malic enzyme Me1	M7WHN9	up	up	2.44	up	A0A0K3CAF6	0.35	down	n.d.	0.49
Acetyl-CoA carboxylase Acc1	M7XLR4	4.25	3.93	n.d.	up	A0A0K3C6V6	up	up	down	down
Fatty acid synthase subunit beta, fungi type Fas1	M7WSW5	3.47	2.29	up	n.d.					
Fatty acid synthase subunit alpha, fungi type Fas2	M7XM89	4.09	2.95	up	up	A0A0K3C4G6	2.76	up	n.d.	down
Glycerol-3-phosphate dehydrogenase (NAD+) Gpd	M7WSY9	n.d.	n.d.	up	n.d.	A0A2S9ZYP9	0.27	0.48	down	down
Glycerol-3-phosphate dehydrogenase (NAD+) Gpd						A0A2T0A892	down	up	up	n.d.
Glycerol-3-phosphate dehydrogenase (NAD+) Gpd						A0A0K3CDD5	0.48	down	up	up
Glycerol-3-phosphate O-acyltransferase Gat1	M7X5G5	2.66	up	n.d.	down	A0A0K3CHG4	down	up	down	4.35
Delta-9 fatty acid desaturase 9FAD	M7XI95	up	up	down	n.d.	A0A191UMV5	n.d.	2.03	up	up
Diacylglycerol O-acyltransferase DGA1	M7WKS9	up	n.d.	n.d.	n.d.	A0A191UMW0	down	3.13	n.d.	n.d.

#### 3.6.6. Effect of Ammonia Supplementation on the Protein Expression

To understand the effect of ammonia on the biomass and lipid formation of IFO0559 and IFO0880, the differences in protein production between samples without supplementation (0 mM) and samples with 0.03% ammonia were analyzed.

The central nitrogen metabolism is composed of glutamate dehydrogenase (Gdh1), glutamine synthase (Gln1/2), and glutamate synthase (Glt1). Under nitrogen-limited conditions, these proteins were upregulated in NP11 [53]. In accordance, elevated levels of Gln2 and Gdh1 were detected in control samples (0 mM) in IFO0559 after 96 h, and lower levels were observed in samples after 144 h (Table 6). While Gdh1 was not detected in IFO0880, Gln 2 and Glt1 were downregulated nearly 2-fold. In contrast, Gln1 was detected in higher amounts for samples with higher nitrogen concentrations in both IFO0559 and IFO0880. The enzyme catalyzes the condensation of glutamate and ammonia to form glutamine.

The urea cycle converts ammonia to urea, in *S. cerevisiae* the urea can afterwards be exported into the medium [68]. In this study, proteins related to the metabolism of the urea cycle were detected in increased amounts, including arginosuccinase, argininosuccinate synthase, arginase, and ornithine carbamoyltransferase in both strains (Table 6). After 96 h, the level of these proteins increased in IFO0880, with the exception of argininosuccinate synthase. Jagtap et al. hypothesized an increased concentration of ornithine, which is produced by arginase in the urea cycle, reflecting the utilization of amino acids for the production of energy in IFO0880 [69]. This might contribute to the better growth of IFO0880 reported in our work. However, expression of carbamoyl-phosphate synthase, the rate-limiting step of the urea cycle, is downregulated in both strains, with the exception of IFO0559 after 144 h. Awad et al. investigated the effect of different nitrogen sources on biomass and lipid production in the oleaginous yeast *Cutaneotrichosporon oleaginosus.* They observed increased biomass as well as lipid accumulation when cells were grown on urea instead of ammonium sulphate as a nitrogen source [70]. As YNB only contains ammonium sulphate, the supplemented ammonium was converted into urea, which could explain the better growth observed in this study, including the higher lipid accumulation.

**Table 6 microorganisms-11-00553-t006:** Differentially expressed proteins involved in ammonia metabolism discussed here. For IFO0559 and IFO0880, fold change of samples grown with ammonia supplementation compared to respective control (0 mM). Protein names and UniProt accession numbers are listed. Abbreviations are written in bold. Further fold change at 96 h and 144 h of samples are stated. Fractions below 1 depict downregulation. n.d. = not detected; up = upregulated below cut-off score; down = downregulated below cut-off score. Cut-off score: significance ≤ 2 or fold change ≤ 2.

	IFO0559			IFO0880		
	Accession	96 h	144 h	Accession	96 h	144 h
Ammonia and Urea Cylce						
Glutamate dehydrogenase (NADP+) Gdh1	M7X2B5	down	up	A0A2T0AGW3	n.d.	n.d.
Glutamine synthetase Gln1	M7XEY4	n.d.	up	A0A0K3C755	up	n.d.
Glutamine synthetase Gln2	M7Y051	0.49	n.d.	A0A0K3CSP6	n.d.	0.44
Glutamate synthase (NADPH/NADH) Glt1	M7WY92	n.d.	n.d.	A0A0K3CF34	down	n.d.
Arginosuccinase	M7X7Q5	up	n.d.	A0A0K3CRK9	up	2.63
Argininosuccinate synthase	M7WMY0	2.08	up	A0A0K3CJ95	up	up
Arginase	M7XGF4	2.17	n.d.	A0A0K3CJ80	3.57	up
Ornithine carbamoyltransferase	M7WUR1	up	n.d.	A0A2T0AE55	2.44	up
Carbamoyl-phosphate synthase (glutamine-hydrolyzing)	M7WUQ8	down	up	A0A0K3C8U1	0.39	0.33

## 4. Conclusions

The results presented in this study provide valuable insight into the adaptation mechanisms of *Rhodosporidium toruloides* to low toxic amounts of Cu(I) and Cu(II). Interestingly, carotenoids were neither significantly increased, nor decreased after the 144 h cultivation period. In contrast, an increase in lipid titers was detected, when upon Cu(II) exposure and proteins related to fatty acid biosynthesis were upregulated under stress conditions. Additionally, proteomic adaptations were observed, indicating detoxification by the involvement of the antioxidant protein machinery and likely by transport of copper into different organelles such as vacuole, mitochondria, Golgi, and lipid droplets. Proteomic data also revealed that copper toxicity in *R. toruloides* is likely not solely a consequence of oxidative stress but also interferes with iron-sulfur and zinc proteins.

## Figures and Tables

**Figure 1 microorganisms-11-00553-f001:**
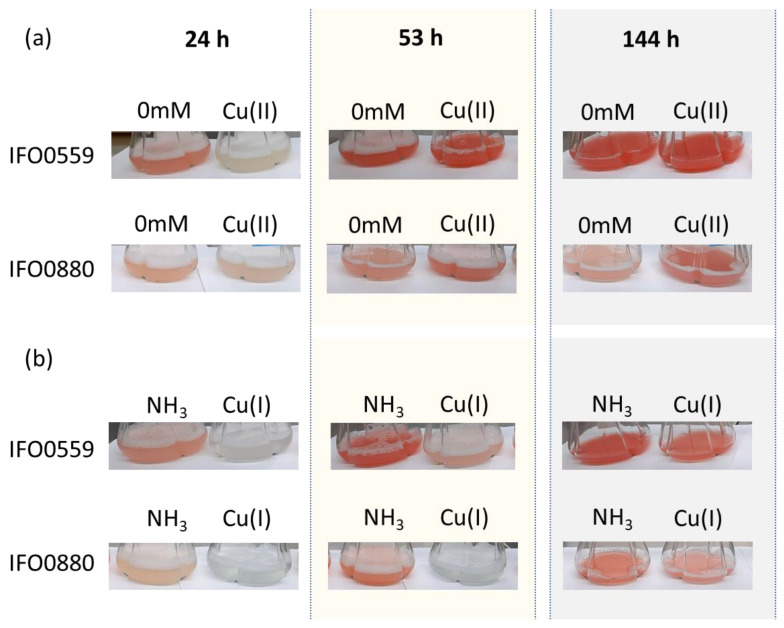
Visible changes in color and turbidity of *Rhodosporidium toruloides* IFO0559 and IFO0880 during shaking flask cultivation. (**a**) Cultivations without supplement (0 mM) versus with 0.5 mM Cu(II). (**b**) Cultivations with 0.03% ammonia (NH_3_) versus with 0.03% ammonia (NH_3_) and 0.5 mM Cu(I).

**Figure 5 microorganisms-11-00553-f005:**
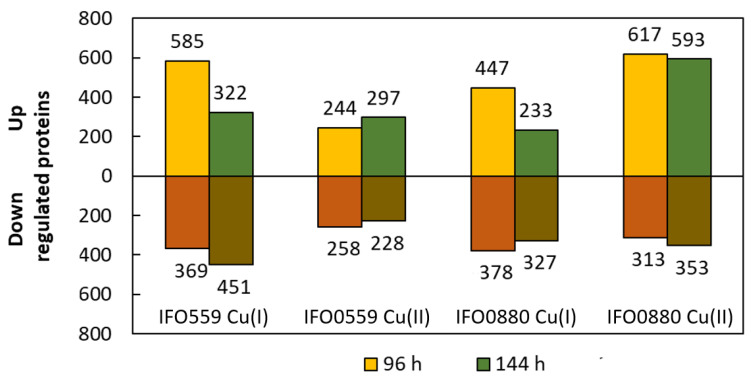
Summary of the proteins quantified with significantly different abundance between samples exposed to 0.5 mM Cu(I) or Cu(II) in comparison to control group, whereas ammonia served as control to Cu(I)-treated samples and 0 mM served as control to Cu(II)-treated samples, at two different time points, 96 h (yellow/red) and 144 h (green/brown). Significance ≥ 2, fold change ≥ 2, detected in at least one sample per group, LFQ by PEAKS Studio.

**Table 1 microorganisms-11-00553-t001:** Differentially expressed proteins involved in oxidative stress discussed here. For IFO0559 and IFO0880, fold change of samples grown under Cu(I) and Cu(II) stress was compared to respective control. Protein name and UniProt accession numbers are listed. Further fold changes at 96 h and 144 h of samples are stated. Fractions below 1 depict downregulation. n.d. = not detected; up = upregulated below cut-off score; down = downregulated below cut-off score. Cut-off score: significance ≤ 2 or fold change ≤ 2.

IFO0559						IFO0880					
		Cu(I)		Cu(II)				Cu(I)		Cu(II)	
	Accession	96 h	144 h	96 h	144 h		Accession	96 h	144 h	96 h	144 h
Catalases, peroxidases and superoxide dismutases											
Catalase	M7XNG2	up	up	0.49	n.d.	Catalase	A0A2T0A1Z3	0.35	down	up	up
Peroxidase	M7XK01	up	down	2.22	2.76	Superoxide dismutase	A0A2T0A5F0	down	down	3.85	3.70
Phosphatidic acid phosphatase type 2/haloperoxidase family protein	M7XV48	n.d.	n.d.	0.47	down	Manganese superoxide dismutase	A0A0K3CNV9	down	n.d.	up	2.13
Glutathione peroxidase	M7WYV4	up	0.46	up	n.d.						
Phosphatidic acid phosphatase type 2/haloperoxidase	M7XF47	n.d.	2.61	n.d.	n.d.						
Redoxins											
Thioredoxin	M7WQ15	n.d.	0.3	2.04	up	thioredoxin	A0A0K3CIV3	down	n.d.	5.88	up
Thioredoxin 1	M7X589	3.23	up	n.d.	up	thioredoxin h	A0A0K3CF59	n.d.	n.d.	10.00	8.33
Thioredoxin family Trp26	M7XEN4	n.d.	0.46	n.d.	down	thioredoxin family Trp26	A0A0K3CFJ6	0.32	down	n.d.	n.d.
Thioredoxin fold domain protein	M7WL87	2.02	2.06	n.d.	n.d.	Thioredoxin-like fold	A0A2T0AFR0	up	2.76	n.d.	n.d.
Thioredoxin-like fold domain protein	M7XWZ6	2.05	down	up	n.d.	Thioredoxin-like_fold domain-containing protei	A0A0K3C742	n.d.	2.09	down	up
Thioredoxin-like protein 5	M7XZL4	down	up	5.00	2.33	Thioredoxin-domain-containing protein	A0A2T0ABL4	n.d.	n.d.	0.41	n.d.
						Thioredoxin domain-containing protein	A0A2T0AIM8	2.02	2.23	6.25	n.d.
						Thioredoxin-like fold	A0A2S9ZX64	up	n.d.	2.94	down
						Thioredoxin-like protein	A0A2T0A3T0	down	n.d.	2.27	down
						Thioredoxin-like [2Fe-2S] ferredoxin-domain containing protein	A0A2T0ADW8	3.33	3.11	0.07	2.08
						putative Phosphoadenylyl-sulfate reductase Thioredoxin	A0A0K3CRW2	n.d.	n.d.	n.d.	5.56
Mitochondrial peroxiredoxin 6 1-Cys peroxiredoxin	M7X0P7	4.32	n.d.	down	up	Mitochondrial peroxiredoxin prx1	A0A2T0A015	6.15	2.42	2.78	3.33
						peroxiredoxin Q/BCP	A0A0K3CD89	n.d.	down	3.33	2.63
Ferredoxin	M7XKB3	down	0.38	n.d.	up						
Adrenodoxin-type ferredoxin	M7XYT4	down	0.15	down	down						
Glutaredoxin 3	M7XKA6	up	down	up	5.07	Glutaredoxin	A0A0K3CPC6	n.d.	n.d.	up	2.70
						Glutaredoxin-1	A0A2T0A9I9	0.43	0.4	n.d.	n.d.
						glutaredoxin domain containing protein	A0A0K3CDE9	n.d.	up	0.31	up
						Redoxin	A0A2T0AE18	up	2.14	n.d.	up
Glutathione											
Glutathione S-transferase	M7WZE5	4.52	up	2.17	up	glutathione S-transferase	A0A0K3C9R2	7.38	5.01	10.00	7.14
Glutathione S-transferase	M7X890	3.23	down	0.40	0.29	glutathione S-transferase	A0A0K3CP32	6.28	9.48	3.13	4.17
Glutamate synthase (NADH)	M7WY92	n.d.	n.d.	6.67	n.d.	Glutathione S-transferase	A0A2T0AH77	3.09	3.94	2.78	3.45
Glutathione S-transferase	M7WRR3	2.09	up	n.d.	up	Glutathione-S-transferase	A0A0K3CPH2	n.d.	up	3.23	2.78
Glutathione S-transferase	M7XMG8	3.57	up	n.d.	up	Glutathione S-transferase kappa	A0A0K3CEA2	up	0.44	2.78	n.d.
Glutathione S-transferase	M7WLE1	n.d.	0.41	down	n.d.	Glutathione-S-transferase	A0A2T0A4G8	n.d.	up	up	2.94
Glutathione S-transferase domain containing protein	M7Y0E4	2.26	n.d.	n.d.	n.d.	Glutathione S-transferase C-terminal-like protein	A0A2T0AI77	up	down	n.d.	2.70
						Glutathione S-transferase C-terminal-like protein	A0A0K3CLG9	up	up	2.22	up
						Glutathione S-transferase C-terminal-like protein	A0A2S9ZZV1	up	up	0.15	up
						Glutathione S-transferase (fragment)	A0A2T0A7J4	2.01	2.32	n.d.	up
						glutathione transferase omega-1	A0A0K3CKG3	n.d.	down	2.13	n.d.
						Glutathione transferase omega-1	A0A2T0AG77	n.d.	12.94	n.d.	n.d.
						Glutamate synthase (NADH)	A0A0K3CF34	5.00	down	2.86	up
						Glutamate--cysteine ligase	A0A0K3CD10	5.22	4.47	3.33	5.88
Glutathione synthetase	M7WY92	n.d.	n.d.	6.67	n.d.	Glutathione synthetase	A0A2T0A1D6	up	n.d.	2.00	up
						S-formylglutathione hydrolas	A0A0K3CIW3	n.d.	down	3.85	9.09
						Lactoylglutathione lyase	A0A061B476	n.d.	up	3.33	2.56
						Glutathione hydrolase	A0A2T0A164	0.49	up	up	5.00
Hydroxyacylglutathione hydrolase	M7XFE4	2.49	n.d.	n.d.	n.d.	Hydroxyacylglutathione hydrolase	A0A2T0ADD4	down	n.d.	up	2.22

## Data Availability

Not applicable.

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
