# Peer review of "Adaptation of Proteome and Metabolism in Different Haplotypes of Rhodosporidium toruloides during Cu(I) and Cu(II) Stress"

_microorganisms, 2023, doi:10.3390/microorganisms11030553_

Round 1

Reviewer 1 Report

In this work, authors performed cell cultures of two Rhodosporidium toruloides haplotypes, namely IFO0559 and IFO0880, in YNB liquid media in the presence of 0.5 mM Cu(I) or Cu(II), and performed routine analysis together with proteomic and metabolic analysis. Solid data were produced and useful information was generated for the community to understand the responses to copper stress by the oleaginous yeast R. toruloides. Therefore, it could be a valuable reference. However, there were major issues to be properly addressed before accepted for publication. Comments,

1. According to Section 2.1, R. toruloides cells were cultivated in YNB liquid media (6.7 g/L Yeast Nitrogen Base, 40 g/L glucose), plus 0.5 mM Cu(II) or 0.5 mM Cu(I) + 0.03% ammonia, when necessary. It was clear that the C-source of the media was low and the C/N molar ratio should also be low such that lipid accumulation failed to initiate, as indicated by cellular lipid contents were lower than or close to 20 wt%. For this reason, fatty acid compositional data in Section 3.5 should be treated with care, because essentially minimal neutral lipids were stored intracellularly, while majority of fatty acid compositional data from oleaginous yeasts found in literature were of neutral lipids from lipid-rich cells.

2. A major flaw of this work was the lack of glucose consumption data for all cultures shown in Figs 2 and 3. Because the initial glucose concentration was 40 g/L glucose, it was likely that glucose was exhausted after 3-4 days. Note that samples for proteomic analysis were obtained after 96 h and 144 h of the culture. If glucose was exhausted, the interpretation of  the proteomic analysis data should be treated with care.   

3. Another flaw of this work was, in fact, without pH regulation/control for all cultures. In  Section 2.1, no information was provided regarding the culture pH. While the presence of Cu could make differences, the variation of culture pH could also make major differences. Please comment.

4. Reference 6 as well as a recent work (PMID: 36394004) demonstrated the capacity of R. toruloides haploid strains for better lipid production with faster glucose consumption rates, however, the current study failed to produce lipids. If authors intended to study copper stress responses with a concurrent lipid production, culture conditions should have been altered. On the other hand, if authors intended to study copper stress responses only, metrics related to lipid production, for example, lipid content, titer and fatty acid composition, should not be emphasized. Authors should clearly state the purpose of this study in Abstract and Introduction.

5. For the culture with Cu(I), an interesting question was to what extent that Cu(I) was oxidized to Cu(II) over time?

Reviewer 2 Report

Reference: Microorganisms 2195127

                Cevelius et al described the impact of the addition of Cu(I) and Cu(II) upon the physiological behavior (mainly the synthesis and the fatty acid composition of cellular lipids, the production of carotenoids and the proteome profile) of two Rhodosporidium toruloides haplotypes (IFO 0559 and IFO 0880) during growth on glucose-based media in shake-flask experiments. The presented results could be an interesting addition in the existing knowledge concerning the production of lipophilic compounds performed by the non-conventional yeast Rhodosporidium toruloides. However, a number of items requests revisions / additions / reconsiderations before any acceptance procedure.

                1) Figs 2a and 3a are of very low quality. Please increase the size of the graphs and the symbols. Please also, instead of the representation of biomass in OD, do the respective representation in g of dry cell weight (g DCW) per L of medium. Please also do the calculation of kinetic parameters (i.e. μmax) for the various trials performed.

                2) Figs 2b and 3b are completely unclear. I consider that we would have a much better appreciation of what is going on into the fermentation medium, if we could see the results that appear in these figs in the form of tables (for t=48h, 96 h and 144 h). In this (these) table (tables) we would also see the consumption of glucose, and we could also see the appearance of total biomass yield on glucose consumed (YX/S, in g/g) and lipid yield on glucose consumed (YP/S, in g/g), therefore a much more global consideration and appreciation concerning the metabolism of the two Rhodosporidium toruloides haplotypes would be presented.

                3) Throughout the submitted m/s, discussions concerning the oleaginous character and potential of Rhodosporidium toruloides are presented. However, the accumulation of storage lipids is not exceptional in all culture conditions imposed. Please do discussions. Authors should be also aware that Rhodosporidium sp. can also produce appreciable quantities of intra-cellular polysaccharides [see i.e.: Proc Biochem (2022) 123, 52–62; Fermentation 2022, 8, 713]. Moreover, according to the results concerning the lipid accumulation in DCW (% w/w) it seems (although it is not clear through figs 2b and 3b that need to be replaced by graphs) that lipids (in % w/w) seem to decrease in some of the trials performed (cellular lipid biodegradation/turnover, in accordance with results reported for yeasts lipid Yarrowia lipolytica and Cryptococcus curvatus, yet not many results concerning Rhodosporidium toruloides exist). Some discussions on this point are requested.

                4) The alteration in the process of accumulation of storage lipids and the FA composition of produced lipids by oleaginous Rhodosporidium toruloides when various “exogenous” compounds (i.e. plant extracts, lignosulfonates, etc) are added vs the control experiments (viz. no addition of compounds) has been already demonstrated in earlier or more recent studies. Please perform discussions and comparisons with the present study. See i.e.: Appl Microbiol Biotechnol. (1985) 22, 41–45; Proc Biochem (2022) 123, 52–62; Fermentation 2022, 8, 713.

                5) Low-aliphatic chain FAs (i.e. C12:0 and C14:0) have been reported (in low concentrations in any case) to exist as compounds in the cellular lipids of Rhodosporidium toruloides. I do not understand why the authors have chosen the FA C12:0 as internal standard in their FAME composition analyses.

                Discussions and comparisons dealing with the international literature on various points of the presented topic are missing. In some figs the presentation of results is indeed problematic. The current m/s, thus, should be re-submitted after major revisions in the points raised by the referee.

Round 2

Reviewer 1 Report

Authors addressed all comments well. It is now acceptable. 

Reviewer 2 Report

In my point of view, in figures 2a and 3a, the authors could increase the size of the symbols used to describe their kinetics. Otherwise, the paper is fine for me. The correction of the graph (if authors want to follow my suggestion) could be done in the status of the correction of proofs. The paper can be accepted.